ns

# Photocatalytic dehydrogenative C-C coupling of acetonitrile to succinonitrile

Xian Zhou[1], Xiaofeng Gao[2], Mingjie Liu[1], Zirui Gao[2], Xuetao Qin[2], Wenhao Xu[1], Shitong Ye[1], Wenhua Zhou[1], Haoan Fan[1], Jing Li[1], Shurui Fan[3], Lei Yang[4], Jie Fu [1], Dequan Xiao [5], Lili Lin [3] ✉, Ding Ma [2] ✉ & Siyu Yao [1,2] ✉

The coupling of acetonitrile into succinonitrile, an important terminal dinitrile for value-added nylon production, via a dehydrogenative route is highly attractive, as it combines the valuable chemical synthesis with the production of green hydrogen energy. Here, we demonstrate that it is possible to achieve a highly selective light driven dehydrogenative coupling of acetonitrile molecules to synthesize succinonitrile using anatase $TiO_2$ based photocatalysts in aqueous medium under mild conditions. Under optimized conditions, the formation rate of succinonitrile reaches 6.55 mmol/($g_{cat}$*h), with over 97.5% selectivity to target product. Mechanism studies reveal that water acts as cocatalyst in the reaction. The excited hole of anatase semiconductor oxidizes water forming hydroxyl radical, which subsequently assists the cleavage of $sp^3$ C-H bond of acetonitrile to generate ·$CH_2CN$ radical for further C-C coupling. The synergy between $TiO_2$ and Pt cocatalyst is important to enhance the succinonitrile selectivity and prevent undesirable over-oxidation and hydrolysis. This work offers an alternative route to prepare succinonitrile based on renewable energy under mild conditions and avoid the use of toxic reagents and stoichiometric oxidative radical initiators.

Succinonitrile (SN), an important $C_4$ intermediate, is mainly used to synthesize 1,4-diaminobutane (DAB) for the production of high-performance polyamides, such as nylon-4,6 (Stanyl®) and nylon-4,10 (EcopaXX®)[1,2]. Due to the wide applications in automobile, electronics and other industries, the annual demand of succinonitrile is growing rapidly[3,4]. Additionally, succinonitrile is used to synthesize paints and electrolyte additives for lithium batteries[5,6].

Succinonitrile is commercially produced by the addition of hydrogen cyanide to acrylonitrile[7,8]. Although high yield is achieved in this atom economic pathway, the highly toxic hydrogen cyanide is used as both reactant and catalyst. Thus, strict safety regulation and hazardous substance control are required during the operation and

post-treatment procedures (Fig. 1a). The dehydration reaction between succinic acid and ammonia produces SN in the presence of solid acid. But the relatively low yield and harsh condition significantly limit its applicability. To overcome these shortcomings, researchers have turned to the SN synthesis based on acetonitrile, a byproduct of acrylonitrile production (Sohio process) and mainly burned out as fuel without further application (releasing large amount of $NO_x$ pollutant). Both anionic coupling assisted with an organometallic complex and radical coupling initiated with radicals from peroxides (Fig. 1b, c) have been proposed to synthesize SN[9,10]. Despite of high chemospecificity, these synthetic routes require the use of stoichiometric amount of expensive butyllithium and peroxide as the hydrogen acceptors of the

[1]Key Laboratory of Biomass Chemical Engineering of Ministry of Education, College of Chemical and Biological Engineering, Zhejiang University, Hangzhou 310027, China. [2]Beijing National Laboratory for Molecular Sciences, College of Chemistry and Molecular Engineering and College of Engineering, Peking University, Beijing, China. [3]Institute of Industrial Catalysis, State Key Laboratory of Green Chemistry Synthesis Technology, College of Chemical Engineering, Zhejiang University of Technology, Hangzhou 310014 Zhejiang, China. [4]Zhejiang Henglan Science and Technology Co. Ltd, Hangzhou 310027, China. [5]Center for Integrative Materials Discovery, Department of Chemistry and Chemical and Biomedical Engineering, University of New Haven, West Haven, CT 06516, USA. ✉e-mail: linll@zjut.edu.cn; dma@pku.edu.cn; yaosiyu@zju.edu.cn

## Industrial succinonitrile synthesis

### a. Nucleophilic addition

**Toxic reagents involved**

## Coupling of acetonitrile

### b. Carbanionic coupling

### c. Oxidative radical coupling

**Complicated procedure, Stoichiometric reagent required**

## *Our Strategy*

### d. Photocatalysis coupling

**Mild condition, High atom economy**

**Fig. 1 | Reaction pathways to synthesize succinonitrile. a** Nucleophilic addition pathway applied in industrial synthesis; (**b**) Carbanionic coupling pathway; (**c**) oxidative radical coupling pathway and **d** the photocatalytic dehydrogenative coupling of acetonitrile proposed in this work.

$sp^3$ bond of $CH_3CN$, which is not desirable for large-scale production. If $CH_3CN$ molecules could be coupled via a catalytic dehydrogenation route, the whole process can be cleaner with even improved atom economy. The generated hydrogen can be used as clean energy. However, due to the thermodynamic limit (Fig. S1), it is impossible to achieve this reaction via traditional thermal catalysis.

The development of photocatalyst provides opportunities to synthesize chemicals in a cleaner manner using renewable solar energy, by overcoming the thermodynamic limit under ambient conditions[11–18]. Using semiconductor-based heterogeneous photocatalysts, facile product separation and catalyst regeneration could be fulfilled[19–24]. As illustrated in Scheme 1d, we aim to synthesize succinonitrile by a light-driven direct dehydrogenation of $CH_3CN$ together with the evolution of high purity hydrogen using semiconductor photocatalysts. In this process, the precise activation of sp³ C-H bonds and keeping -CN intact are the prerequisites.

Here, we report a one-step synthesis of succinonitrile from $CH_3CN$ via light driven C-H bond cleavage and subsequent C-C dehydrogenation. By the $Pt/TiO_2$ (anatase) photocatalyst, the formation rate of succinonitrile reached 6.55 mmol/($g_{cat}$*h) with >97.5% selectivity. Mechanism studies revealed that the photo-induced holes oxidized water to form hydroxyl radicals. The radical chain transfer process activated the C-H bond and formed •$CH_2CN$ radicals. The coupling of two •$CH_2CN$ radicals generated succinonitrile. While the excited electron reduced water to produce hydrogen. The synergistic effect between anatase $TiO_2$ and Pt played an important role for the efficient

and selective SN formation. The deep trapping states over anatase surface effectively accelerated the formation of hydroxyl radicals to enhance the local concentration of •$CH_2CN$ radicals, which benefited the coupling reaction. Pt NPs accelerated the consumption of $H^+$ via hydrogen evolution, preventing undesirable side reactions of acetonitrile hydrolysis.

## Results and discussions

### Synthesis and characterization of photocatalysts

All the Pt or other metal based photocatalysts were prepared using the photo-deposition[25] of metal precursors on commercial anatase, rutile or P25 titanium oxides. The resulting catalysts were denoted as M/$TiO_2$- A, -R or -P. The XRD patterns of as-prepared $Pt/TiO_2$ catalysts indicated that all the loaded metals were finely dispersed on the $TiO_2$ surface without agglomeration (Fig. 2a). The Pt $L_3$ edge XANES spectra of $Pt/TiO_2$-A, $Pt/TiO_2$-P and $Pt/TiO_2$-R in Fig. 2b demonstrated that the loaded Pt species were fully reduced. The fitting results of EXAFS spectra (Table S1) showed that the Pt-Pt first shell coordination numbers of each of the catalysts were in the range of 8.9-9.9, indicating the average particle sizes of around 3 nm (Fig. 2c)[26,27]. The high angle abbreviation dark field scanning transmission electron microscope (HAADF-STEM) images showed that the photo-deposited Pt NPs were in irregular shape (Fig. 2d−f), with an average Pt size of 3 nm. The UV-Vis spectroscopy (Fig. S2) showed that the bandgaps of photocatalysts were 3.20 eV for $Pt/TiO_2$-A, 2.95 eV for $Pt/TiO_2$-P, and 2.98 eV for $Pt/TiO_2$-R. The photocurrent density (Fig. S3) decreased in the following

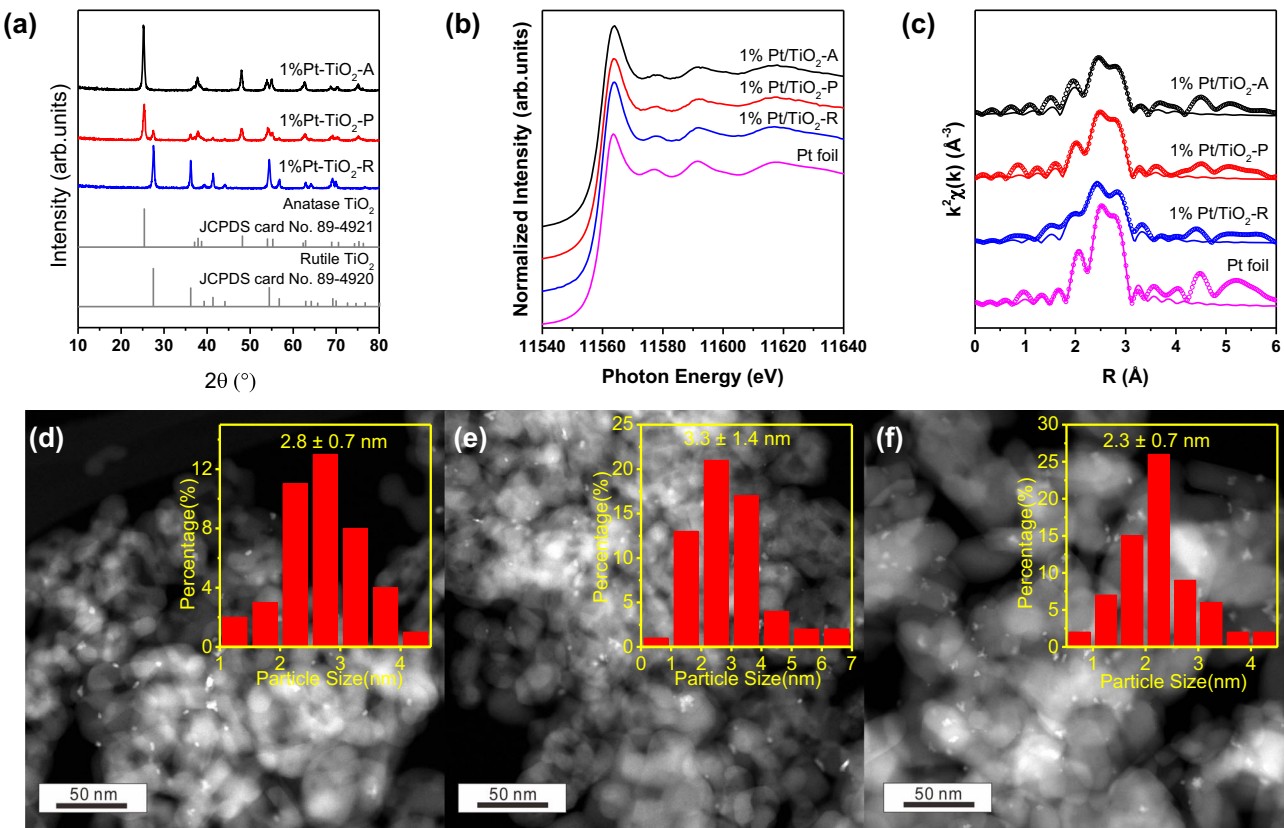

**Fig. 2 | Structure characterization of Pt/TiO₂ photocatalysts. a** X-ray diffraction pattern of 1% Pt/TiO₂ photocatalysts; (**b**) Pt L₃ edge XANES and EXAFS spectra of 1% Pt/TiO₂-A, -P and -R catalysts together with Pt foil reference; (**c**) the fitting EXAFS results; HAADF-STEM images of 1% Pt/TiO₂-A (**d**), -P (**e**) and -R (**f**) catalysts. The particle size distribution of Pt of each catalyst is shown in the insert figures.

order: Pt/TiO₂-A > Pt/TiO₂-P > Pt/TiO₂-R, suggesting that the light harvesting ability of catalysts increased with the percentage of anatase phase in the support.

**Photocatalytic dehydrogenative coupling of acetonitrile**

The photocatalytic reactions were performed in 20 mL parallel batch reactors (Fig. S4) using aqueous solution of $CH_3CN$ as both reaction medium and reactant. In a typical test, 20 mg fine powder of catalyst was placed into the system, working at 60 °C for two hours. The 365 nm 10 W LED light was used as the light source. In Fig. 3a, no reaction occurred in the absence of light radiation, suggesting that acetonitrile was stable in aqueous solution at 60 °C. Only a small amount of $CO_2$ (0.31 mmol/($g_{cat}$*h)) was produced over TiO₂-P, which can be attributed to the overoxidation of $CH_3CN$ on the bare support (Table S2). No coupling products appeared under the catalysis of pure TiO₂ semiconductors. The loading of transition metal cocatalysts had significant influence on the catalytic activity and product distribution (Table S2). When Pt was introduced onto the catalyst, 1% Pt/TiO₂-R exhibited weak $H_2$ evolution together with the formation of $CO_2$ as the only carbon-containing product. In contrast, a high activity of SN at 6.55 mmol/(g*h) was achieved by the 1% Pt/TiO₂-A with a selectivity of 97.5% and quantum yield of 4.4%. Only a trace amount of $CO_2$ (1.4%) and acetamide (1.1%) were detected, which were the by-products from overoxidation and acetonitrile hydrolysis. Simultaneously, about 9.76 mmol/($g_{cat}$*h) $H_2$ was produced, suggesting that the SN was obtained via dehydrogenative coupling. The Pt/TiO₂-P with >75% anatase phase exhibited good activity to the SN production (4.97 mmol/(g*h)), which is 75% of the activity of Pt/TiO₂-A. Other semiconductors supported Pt photocatalysts, including Ti based perovskites, ZnO, BiOCl, CdS and $C_3N_4$, were evaluated for the same reaction under similar conditions to that of the Pt/TiO₂ catalysts. Only Pt/BiOCl and Pt/

$Bi_2MoO_6$ showed weak activity (0.22 and 0.16 mmol/(g*h)) for the $CH_3CN$ coupling (Fig. 3a and Table S3). These results suggested that the activity of dehydrogenative coupling of acetonitrile strongly depended on the choice of semiconductors. The anatase phase TiO₂ exhibited exclusive advantages for SN formation. Furthermore, the function of different transition metal cocatalysts was investigated (Fig. 3b). Among all the M/TiO₂-A catalysts, Pt/TiO₂-A showed the highest SN activity, which was 5 times higher than that of Au or Rh/TiO₂-A and more than one order higher than that of Pd, Ag or Fe/TiO₂-A. Moreover, Pt/TiO₂ had the highest selectivity towards SN. The SN selectivity over Au and Rh/TiO₂-A catalysts were about 94%. In contrast, the 13% and 26% of consumed $CH_3CN$ was transformed into the overoxidized product $CO_2$ by Pd and Fe/TiO₂, respectively. Pd and Ag/TiO₂ exhibited 12.2% and 97.7% selectivity towards acetamide (AM), respectively. For Ag/TiO₂-A, the AM formation rate reached as high as 32.0 mmol/g/h, indicating that it is highly active for the hydrolysis side reaction.

The ratio of water and acetonitrile is an important factor for SN synthesis (Fig. 3c). Without water, SN was merely observed. Only when $H_2O$ and $CH_3CN$ were both presented, considerable amount of SN was generated with a maximum rate at $CH_3CN/H_2O$ = 7:3, suggesting that water was probably involved in the dehydrogenative coupling reaction. The change of substrate/catalyst ratio (Fig. 3d) showed that the mass specific activity of Pt/TiO₂-A reduced sharply with the increasing substrate/catalyst (S/C) ratio. When S/C = 1100:1, the coupling rate was 2.65 mmol/(g*h), only 40% of that under the optimal condition. Thus, the dilution of catalyst with excessive amount of reactant had negative effect on the reaction. The coupling reaction may not occur fully on the surface of photocatalyst. Otherwise, the coupling rate should not have changed with a reduced amount of catalyst. The time-dependent reaction rate from 0 to 6 h is presented in Fig. 3e. The accumulated

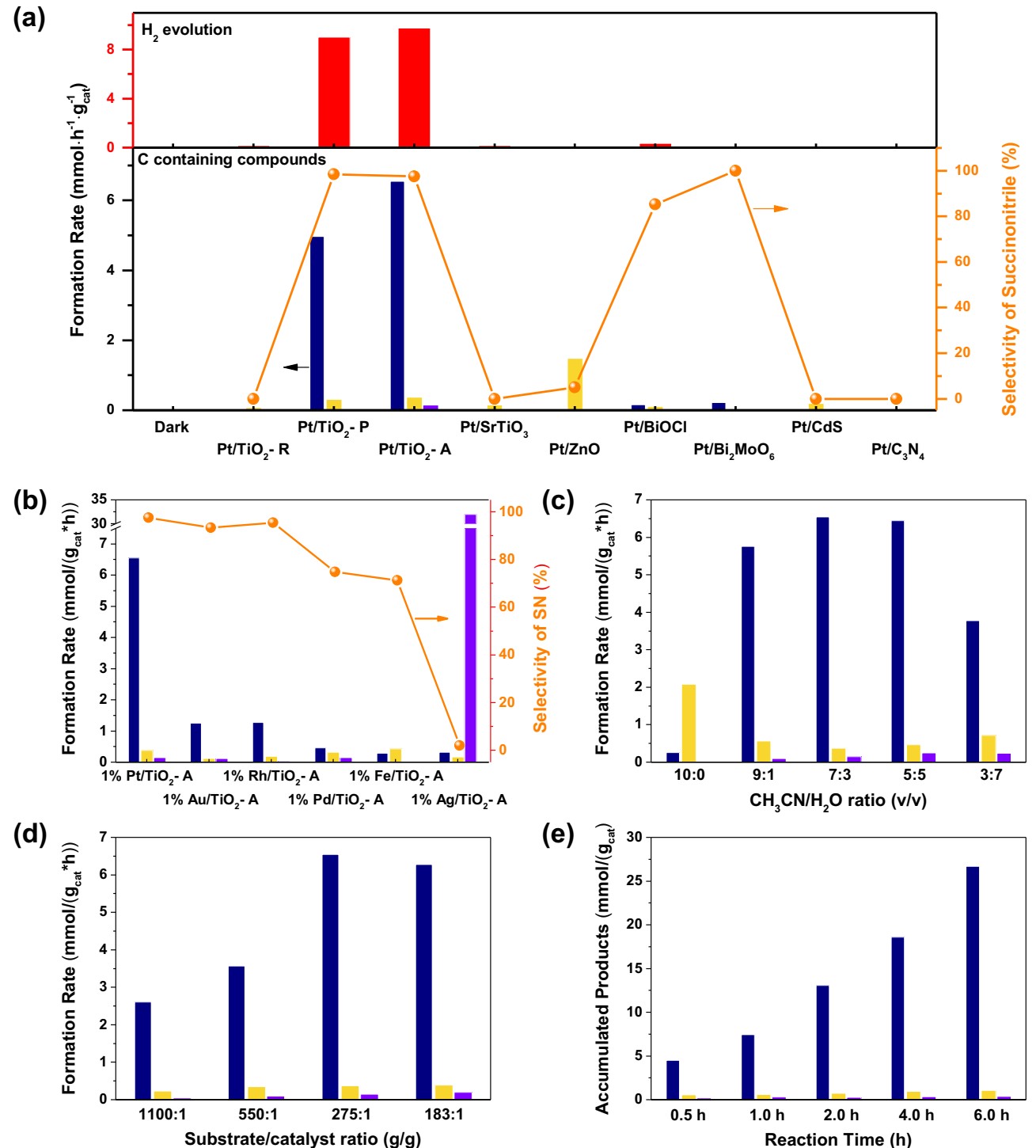

**Fig. 3 | Catalytic performances evaluations of the photocatalytic dehydrogenative coupling reaction over Pt/TiO₂-A catalyst. a** Succinonitrile formation rates over various semiconductor supported Pt photocatalysts; (**b**) transition metal cocatalyst effect on the succinonitrile formation rate and selectivity; influence of (**c**) $CH_3CN$/Water ratio, (**d**) substrate/catalyst ratio on the catalytic performances, and (**e**) time dependent catalytic performance. Reaction condition: 20 mg catalyst, 10 mL aqueous solution of $CH_3CN$ ($CH_3CN$: $H_2O$ = 7:3) with Ar protection, 10 W LED lamp ($\lambda$ = 365 nm), 60 °C, 2 h.

product in 6 h was estimated to 26 mmol/$g_{cat}$, equivalent to a turnover number (TTN) of 570 per Pt atom in 6 hrs.

## Mechanism investigation of the dehydrogenative coupling of acetonitrile

Detailed reaction mechanism of the acetonitrile dehydrogenative coupling was studied. The coupling reaction was a photocatalytic process which occurred in the presence of water. To identify the role of photo-excited holes, electrons and radical intermediates, mechanism studies employing scavengers were performed (Fig. 4a). When $Na_2S$ was introduced into the system as the hole scavenger, the formation of SN was completely ceased. When $AgNO_3$ was added as electron scavenger, only small amount of SN formed in 2 h (0.17 mmol/($g_{cat}$*h)). Simultaneously, significant amounts of overoxidation product $CO_2$ and hydrolysis byproduct acetic acid formed with rates of 15.56 and 9.75 mmol/(g*h), respectively. This result suggested that the

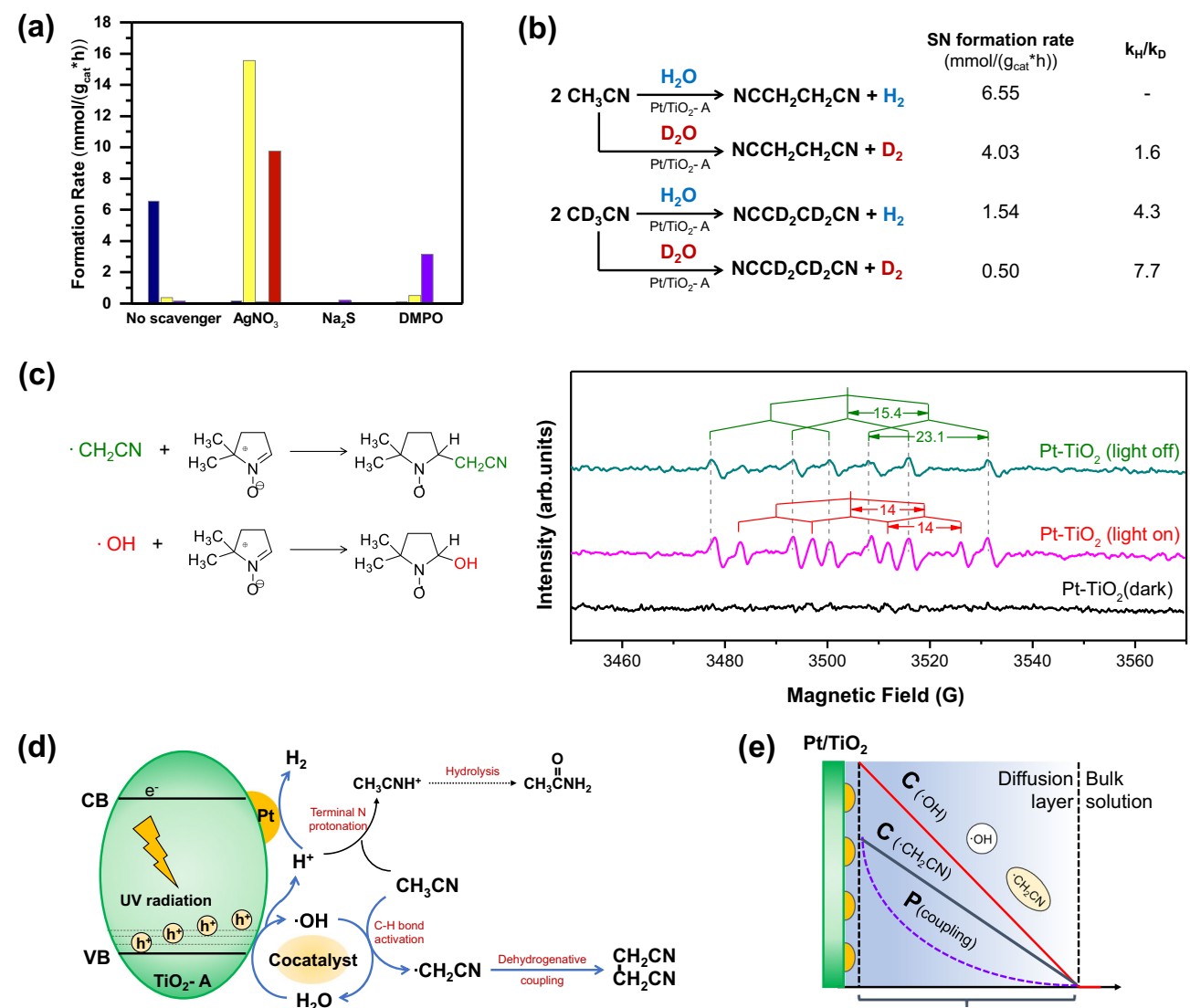

**Fig. 4 | Mechanism study of CH₃CN dehydrogenative coupling over Pt/TiO₂-A catalyst. a** Control experiments of CH₃CN coupling reaction in the presence of electron, hole and radical scavengers; (**b**) determination of C-H(D) and O-H(D) kinetic isotope effect using D₂O and CD₃CN as substitution reagents; (**c**) EPR patterns of radical intermediates trapped using DMPO; (**d**) Proposed reaction mechanism of acetonitrile dehydrogenative coupling and hydrolysis side reaction; and (**e**) Schematic illustration of the reaction region based on •OH diffusion layer assumption.

accumulation of holes on the catalysts tended to promote the oxidation of CH₃CN to CO₂. Meanwhile, the proton formed in the water oxidation reaction by the hole could protonate the terminal N of cyanide, which facilitated the acetonitrile hydrolysis side reaction. The AM byproduct in the normal catalytic reaction was generated via proton-assisted hydrolysis. Therefore, the prevention of accumulation of holes and protons from the water oxidation reaction are highly important to reduce or eliminate the side reactions. Furthermore, dimethyl pyridine N-oxide (DMPO) was added as radical scavenger. A strong deactivation of catalysts was observed (SN formation rate was only 1/550 of the reference test), indicating that the radicals were probably the key intermediates in the CH₃CN coupling reaction. Kinetic isotope effect experiments suggested that both the O-H and C-H dissociation steps had significant influence on the reaction rate (Fig. 4b), suggesting that the water actually participated in the dehydrogenation coupling reaction. The hydrogen atom in the H₂ originated from water rather than CH₃CN. Therefore, water acted as a cocatalyst during the dehydrogenative coupling reaction.

To investigate reaction pathways and identify major radical intermediates, electron spin resonance spectroscopy (EPR) assisted by radical trap reactions were performed using DMPO as trapping reagent (Fig. 4c). In the presence of UV radiation and catalysts, both the signals of trapped •OH and •CH₂CN radicals (marked as red and green respectively) were observed[28]. In comparison, no radical was trapped without light. When radiation was removed after the system reached steady state, an instant consumption of hydroxyl radical occurred. The trapped •CH₂CN was stable, and the •OH radical was a more active radical that could extract H from the sp³ hybridized CH₃- of acetonitrile. Meanwhile, the intensity of hydroxyl radical signals was relatively weaker in the presence of CH₃CN than in pure water (Fig. S5), indicating that the •OH was probably the primary radical species generated from the single electron water oxidation process. The •CH₂CN radical was generated from CH₃CN via a selective α C-H bond cleavage process initiated by •OH. The SN was a product of the coupling reaction between two •CH₂CN. Combining with the KIE results ($k_H/k_D = 4.3$ for CH₃CN/CD₃CN), we deduced that the radical propagation from •OH to •CH₂CN was the rate-limiting step.

Based on the results of mechanism studies, the preferential dehydrogenative coupling reaction of acetonitrile to succinonitrile is a light-driven catalytic process with free radicals as key intermediates

(Fig. 4d). In the photocatalytic reaction, the hole oxidized water to deliver hydroxyl radical and H[+]. During this process, the anatase phase $TiO_2$ that possesses abundant deep trapping states above the valence band, effectively reduced the formation potential required for •OH radical formation[29]. The surface Ti-O bonding stabilized the O[-], which can be further transformed into •OH radicals in aqueous medium (confirmed by the photoluminescence spectroscopy, Fig. S8)[30,31]. Due to the highly efficient •OH formation by anatase, a considerable high concentration of •OH was formed within the diffusive layers, even though the life-time of hydroxyl radical was quite short (the formation of free •OH radicals in the solution was confirmed by the fluorescence probe method, Fig. S9)[32]. In comparison, the rutile phase $TiO_2$ have been widely studied as a material that is inefficient for the •OH radical formation in water. The surface O[-] tended to form >Ti-O-O-Ti< bridge rather than release as radicals[33]. The difference in hydroxyl radical formation rates accounts for the support effect in the $CH_3CN$ coupling reaction. In the subsequent step, the •OH radical selectively activated the α C-H bond, generating the •$CH_2CN$ radicals as the coupling intermediate. The competitive reactions of C-H cleavage and -CN protonation by H[+] generated in the water oxidation step was the important factor determining the reaction selectivity. If the N terminal of cyanide group was protonated, the carbon atom of -CN could be more vulnerable to the attack of water[34]. Acetamide and acetic acid byproducts could be formed subsequently. To prevent the hydrolysis process, efficient hydrogen evolution reaction over the transition metal cocatalyst is crucial for H[+] removal. Those noble metal centers with good $H_2$ evolution abilities (e.g., Pt and Rh) were able to consume the excessive H[+] and suppressed the hydrolysis selectivity[35,36]. Fe and Ag were less efficient for $H_2$ evolution exhibiting high side reaction selectivity[37]. Hence, the synergy between Pt and $TiO_2$ is important to achieve a high yield of SN. Except for the photocatalyst design, the control of radical diffusion is highly important (Fig. 4e). As the bond dissociation energy (BDE) of water is large (497 kJ/mol), the •OH diffusion layer can be very thin[38]. The radical transfer from hydroxyl to •$CH_2CN$ can only occur in the diffusion layer, and the coupling possibility of two •$CH_2CN$ radicals (second order kinetics) can be much smaller in the outer region of the layer than in the vicinity of catalyst surface (where radical concentration is much higher)[39]. Therefore, the majority of products should be generated from the catalyst surface. We anticipate that methods that can significantly enhance the steady state concentration of •OH and •$CH_2CN$ can significantly benefit the reaction rate and yield.

In summary, we demonstrated that succinonitrile could be prepared with high selectivity and considerable rate via a UV light driven dehydrogenative coupling reaction of acetonitrile. Anatase $TiO_2$ supported Pt NPs were identified as the efficient catalytic system, exhibiting a high SN formation rate of 6.55 mmol/(g*h) with over 97.5% of selectivity and 4.4% of quantum yield. Mechanism studies suggested that the reaction proceeded via a hydroxyl radical initiated radical pathway. The synergy between the anatase $TiO_2$ semiconductor with the loaded Pt cocatalyst enhanced the rate of •OH radical formation, radical propagation, coupling and hydrogen evolution, leading to the good performance of catalysts. This work paves a new way towards transforming low-value organic molecules to value-added terminal biofunctionalized compounds in a green, nontoxic and atomic efficient manner.

## Methods
### Materials
All chemicals were used directly without further purification. Anatase $TiO_2$ ($TiO_2$-A) and rutile $TiO_2$($TiO_2$-R) were purchased from Aladdin. $TiO_2$(P25) ($TiO_2$-P), which contained 25% rutile and 75% anatase, was purchased from Degussa. $H_2PtCl_6\cdot6H_2O$, $HAuCl_4\cdot4H_2O$, $PdCl_2$, $RhCl_3\cdot3H_2O$, $AgNO_3$, and $Fe(NO_3)_3\cdot9H_2O$ were purchased from Macklin.

### Catalyst preparation
The M/$TiO_2$ catalysts were prepared by photodeposition method. In the photodeposition (PD) of metal over $TiO_2$ support, 200 mg $TiO_2$ powder and metal precursor ($H_2PtCl_6\cdot6H_2O$ / $HAuCl_4\cdot4H_2O$ / $PdCl_2$ / $RhCl_3\cdot3H_2O$ / $AgNO_3$) were dispersed in 15 mL 20% methanol aqueous solution (v/v) in a quartz flask. Subsequently, the flask was placed in the 365 nm photo-radiation generated by a 1 W LED lamp under argon protection with vigorous stirring for 3 h. After the PD procedure, the M/$TiO_2$ was separated from the suspension by centrifugation, washed 3 times using DI water and dried at 333 K for 12 h before use.

The other Pt modified semiconductor catalysts were prepared using similar procedure as the M/$TiO_2$ photocatalysts.

### Sample characterizations
The photocatalysts were characterized by ICP-AES, X-ray diffraction, X-ray absorption spectroscopic, transmission electron microscopy, EXAFS spectroscopy, ESR spectroscopy, UV-Vis spectroscopic, photoluminescence spectroscopy, and photoelectrochemical measurements were used to study the structure and photocatalytic properties of the catalysts. The experimental details are described in Supplementary Information.

### Photocatalytic reaction test
Photocatalytic experiments (acetonitrile dehydrogenative coupling to succinonitrile) were performed in a top-irradiation Pyrex flask. A 10 W LED light (wavelength 365 nm) (PLS-SXE300, Beijing Trusttech Co., Ltd.) was used as the light source. Typically, 20 mg photocatalysts were dispersed in 10 mL 70% volume acetonitrile aqueous solution under magnetic stirring. Prior to the irradiation, the reaction mixture was deaerated repeatedly with Ar gas for 5 times to thoroughly remove air and dissolved oxygen. During the reaction, the photocatalytic reaction system was kept at 60 °C. To evaluate the photocatalytic hydrogen production and analyze other gas products, the gas-phase composition of the photocatalytic reactor was analyzed by an Agilent 8860 gas chromatograph equipped with 5 Å molecular sieves and HP-Plot columns and thermal conductivity cell (TCD) detector. Liquid products were analyzed by Agilent 8860 gas chromatograph equipped with a column of SH-1 with flame ionization detector (FID).

## Data availability
The data that support the plots within this paper and other finding of this study are available from the corresponding author upon reasonable request.

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

## Acknowledgements

This work is financially supported by the National Key R&D Program of China (2021YFC2101800), Natural Science Foundation of China (22178302, 22002140), Zhejiang Provincial Natural Science Foundation of China under Grant (LR21B030001, LR22B030003), Beijing National Laboratory for Molecular Sciences (BNLMS202003), ZJU-Hengyi Global Innovation Research Center (ZJUHY-B20210203) and Young Elite Scientist Sponsorship Program by CAST (2019QNRC001).

## Author contributions

S.Yao and D.M. and L.L. proposed the project and designed the experiment. X.Z. performed most of the experiments, analyzed the data and prepared the manuscript. X.G., H.F., W.Z. and J.L. carried out the part of catalytic performance evaluation and GC analysis. M.L. carried out the measurement and analysis of transient photocurrent. X.Q. performed the in-situ XAFS characterization. Z.G. carried out the TEM characterization. S.F., W.X and S.Ye prepared the catalysts and evaluated the catalytic performances. L.Y. helped to improv the photocatalytic reactor. S.Yao and D.M. and L.L. revised the manuscript. J.F. and D.X. modified the style of manuscript.

## Competing interests

The authors declare no competing interests.
