## [Peer Review File · Nature Communications]

Photocatalytic Dehydrogenative C-C Coupling of Acetonitrile to SuccinonitrileREVIEWER COMMENTS

Reviewer #1 (Remarks to the Author):

In this work, the authors nicely demonstrated that succinonitrile, an important industrial intermediate for value-added nylon production, could be sustainably achieved via by photocatalytic coupling of acetonitrile using anatase TiO₂ based photocatalysts under mild conditions, and the formation rate of succinonitrile reaches 6.55 mmol/(gcat*h) with over 97.5% selectivity. The authors also performed very carefully experiments to probe the reaction mechanism, which suggests that hydroxyl radical initiated radical pathway. However, there still several critical concerns need to be carefully addressed. Thus, the manuscript could be recommended for publications in Nature Communications only after major revision.

1. One of the major selling points of this manuscript lies in that they provide a mild and promising approach for the production of industrially importance product. Thus, carrying out a large-scale experiment (i. e. using 0.5 to 1 gram of catalyst) or figure out a flow-reactor is important, which can help to evaluate the industrial potential of this method and better support their claims.
2. In this manuscript, only one substrate, acetonitrile, was used in the photocatalytic coupling. From academic viewpoints, it is also very interesting to see whether the mild strategy could be applied for coupling of other types of alkyl nitriles or benzyl nitriles.
3. From the isotopic experiments, it is clear that water is actually consumed to release H₂/D₂, and the amount of H₂ is even more than the succinonitrile product. Then, is still proper to define water as co-catalyst?
4. The stability of photocatalysts is very important parameter for photocatalytic process. The characterizations before and after reaction should be carefully studied. Multi-run experiments or long-term stability of this photocatalyst should be carefully evaluated.
5. In Figure S2, the determination of band width is inaccurate.
6. The time-resolved fluorescence decay spectra of various Pt/TiO₂ should be provided.
9. The quality of the manuscript should be carefully improved. For example, the figure is not consistent with the text (please ref to, line 122 (where Figure 1A should be changed to Figure 2A), line 126 and line 134.), and the reference format is also incorrect; The format for "mL" should be consistent, please refer to line 103 and 151.

Reviewer #2 (Remarks to the Author):

Dear authors,

this is an interesting manuscript that reports a photocatalytic conversion of Acetonitrile to Succinonitrile over titania-supported Pt catalysts. The work was carefully conducted and the manuscript is well written. However, it seems that this manuscript is not appealing enough to the broad readership of Nature Communications because the Succinonitrile market is small. It is not acceptable to claim that "the annual demand of succinonitrile is growing rapidly" based on a resource that was published 15 years back (ref.3, Applied Microbiology and Biotechnology 2007, 75 (4), 751-762.). This reviewer recommends the authors to redirect their work to some of the chemical engineering journals.

Reviewer #3 (Remarks to the Author):

This article describes the photochemical formation of cyanomethyl radicals from acetonitrile with subsequent coupling to make succinonitrile. The system is remarkably efficient, using platinum titanium dioxide as the photocatalyst and producing H₂ as byproduct. Of interest, only the anatase phase of TiO₂ was found to be good for the reaction producing succinonitrile with high selectivity. Rutile was ineffective, and a mixed phase material was less effective for the coupling. I recommend publication after consideration of the following points.

- (1) In figure 2D, I think the reaction is limited by photons, not the amount of catalyst. Can some estimate be made of the quantum efficiency of the reaction. i.e., moles of product/mole of photons?
- (2) p.6, top, succinonitrile is produced at a rate of 6.55 mmol/g/h, whereas H₂ is produced at a rate of 9.76 mmol/g/h. These rates should be equal. Why/how is the excess of hydrogen produced?
- (3) the mechanistic studies of this heterogeneous system are suggestive, but not proof, of that mechanism involves CH₂CN radicals. The observation of spin-trapped radicals is perhaps the best indication of CH₂CN radicals. However, the isotope effect studies are not what I expected. Is there any indication of why the isotope effects (kH/kD) are not multiplicative? (1.6 x 4.3 = 6.9, not 13.1)
- (4) in the SI, p.5, how was the number of catalyst active centers determined in Eq 2?
- (5) in the SI, p.16, Table S4. Why does the rate fall off with reaction time? This is seen for SN, H₂ CO₂ and AM. It would be better to give amounts (in mmol) of products, not just rates which are presumably derived from the observed amounts.
- (6) Other corrections:
 - p.1, line 23 should be "Under optimized conditions"
 - p.2, line 1 should be "under mild conditions"
 - p.2, line 2 from bottom should be "in this atom economic pathway, the highly toxic hydrogen cyanide is used"
 - p.3, line 50 should be "assisted with an organometallic complex"
 - p.3, line 54 should be "of the sp³ bond"
 - p.3, line 55 should be "large-scale production."
 - p.3, line 61 should be "under ambient conditions."
 - p.4, line 81 should be "The resulting catalysts"
 - p.4, line 86 should be "numbers of each of the catalysts"
 - p.4, line 5, Fig. 1 should be "The particle size distribution"
 - p.5, line 104 should be "reaction medium and reactant."
 - p.6, line 130 should be "CH₃CN was transformed"
 - p.7, line 143 should be "have changed with a reduced amount of catalyst."
 - p.8, line 159 should be "When AgNO₃ was added"
 - p.8, line 161 should be "significant amounts of"
 - p.8, line 162 should be "with rates of 15.56 and 9.75 mmol/(g*h), respectively."
 - p.8, line 172 should be "Kinetic isotope effect experiments suggested"
 - p.8, line 174 should be "The hydrogen atom in the H₂ originated from" also, I would say that water is acting as a promoter, not a co-catalyst.
 - p.8, line 178 should be " assisted by radical trap reactions were performed"
 - p.10, line 212 should be "evolution, exhibiting high side reaction selectivity."

Response to the referees:

We thank the referees for the insightful and constructive comments and suggestions in order to help us further improve the manuscript. We have addressed all the comments point-by-point and revised the manuscript accordingly. In this response letter, comments from the referees are summarized in black typeface with the original comments quoted in *italic*, and our responses are in blue typeface. All major changes have been highlighted in blue in the main text. A detailed list of changes in the manuscript is also provided.

A list of main changes we have made:

1. The photoluminescence spectroscopic and fluorescence probe study of the Pt/TiO₂ catalysts were performed.
2. The quantum yield and stability data of the Pt/TiO₂-A catalyst was updated in the manuscript.
3. The kinetic isotope effect experiment was examined and performed for another three times to obtain reliable results.
4. Detailed descriptions of the experimental procedures were provided.
5. More references were added in the revised manuscript.

A point-to-point response to the comments

Referee #1 The reviewer 1 confirms the significance of our research. However, the manuscript could be recommended for publications in Nature Communications only after major revision.

Comment 1. *“One of the major selling points of this manuscript lies in that they provide a mild and promising approach for the production of industrially importance product. Thus, carrying out a large-scale experiment (i. e. using 0.5 to 1 gram of catalyst) or figure out a flow-reactor is important, which can help to evaluate the industrial potential of this method and better support their claims.”*

Reply: We appreciate the reviewer’s positive recommendation and valuable comments. We have seriously considered these and revised our manuscript accordingly. We tried to scale up the reaction. However, due to the limited condition of our lab, we cannot scale up to such a large extent. We performed the reaction in a 100 ml photocatalytic autoclave with the Al₂O₃ crystal window at the bottom of the reactor. 0.1 gram of catalyst was used for the reaction. The UV radiation was provided by a 300 W Hg lamp. The formation rate of succinonitrile was 1.85 mmol/(gcat*h). The mass specific activity of the catalyst dropped to 2/7 of the small-scale reaction due to the different radiation source and other engineering problems (mass transfer probably). Even though, strong scale up effect exists in the system, the production capacity of SN in a single run increased by 40%. We are still working on the optimization of the operation condition and try to scale up the system.

Figure R1. High pressure photocatalytic reactor (equipped with 300w Hg lamp)

Comment 2. “In this manuscript, only one substrate, acetonitrile, was used in the photocatalytic coupling. From academic viewpoints, it is also very interesting to see whether the mild strategy could be applied for coupling of other types of alkyl nitriles or benzyl nitriles.”

Reply: This is a very good question and we have tried to expand the possible substrates for this coupling reaction. The n-butyronitrile, isobutylnitrile and chlorine substituted acetonitrile were tested for the reaction under the same condition with the acetonitrile. The GC signals were listed in Table R1. Severe side reactions occurred in the C₄ nitrile coupling reactions. Only trace amount of dinitrile product was observed, suggesting

there were multiple vulnerable sites in the alkyl for the attack of $\cdot\text{OH}$. No product has been discovered for the di-chlorine substituted acetonitrile photocatalytic coupling reaction in the GC profiles. Therefore, the coupling reactions are limited to the compounds with only $\alpha\text{-C-H}$ bonds over TiO_2 based catalysts. Indeed, we managed to obtain considerable yield of terminal bifunctional compound in the reaction of acetone coupling to 2,5-hexanedione (4.3 mmol/($g_{\text{cat}} \cdot \text{h}$)), methyl acetate coupling to 1,2-ethanediol diacetate (0.9 mmol/($g_{\text{cat}} \cdot \text{h}$)) and methanol coupling to ethylene glycol (1.8 mmol) over Pt/ $\text{TiO}_2\text{-A}$ catalyst.

Table R1 GCMS diagram of other coupled products

Reactant	Production	GCMS
n -Butyronitrile		Isobutyronitrile	None	Dichloro acetonitrile	None	Acetone		
Methyl acetate		Methanol		
Comment 3. “From the isotopic experiments, it is clear that water is actually consumed to release H_2/D_2 , and the amount of H_2 is even more than the succinonitrile product. Then, is still proper to define water as co-catalyst?”

Reply: The H_2 generated is higher than the succinonitrile due to the side reactions, such as the overoxidation of acetonitrile. The active oxygen species including the $\cdot OH$ were the oxidative reactant for the side reactions. For the coupling reaction, water is the co-catalyst. Without water, the coupling reaction cannot occur. Therefore, we believe it is still proper to claim water is the cocatalysts.

Comment 4. “The stability of photocatalysts is very important parameter for photocatalytic process. The characterizations before and after reaction should be carefully studied. Multi-run experiments or long-term stability of this photocatalyst should be carefully evaluated.”

Reply: The stability of the catalyst was not very good. As shown in Table S4, the reaction activity kept dropping with extended reaction time and repeated cycle. Although the structure of the catalyst did not change significantly before and after reaction (Figure R2 and R3), the reasons for the deactivation is still under investigation. One possible reason is the CN^- leaching problem. CN^- was generated in trace amount as the byproduct of photocatalytic reaction and accumulated with radiation time. These species tended to leach the noble metals and damage the structure of metal-support

interface. We are now seeking potential method to enhance the stability of Pt/TiO₂ catalyst for the acetonitrile coupling reaction.

Figure R2. X-ray diffraction patterns of 1%Pt/TiO₂

Figure R3. XPS profiles of Ti 2p, O 1s and Pt 4f in the 1%Pt/TiO₂

Comment 5. “In Figure S2, the determination of band width is inaccurate.”

Reply: The obtained UV-vis diffuse reflectance spectra were first transformed to absorption spectra according to the Kubelka-Munk function¹,

$$F(R) = \frac{(1-R)^2}{2R} \times 100\%$$

where R was the relative reflectance of samples with infinite thickness compared to the reference. Moreover, the band gaps of samples were estimated on the basis of the Tauc equation,

$$F(R)hv = A(hv - E_g)^{\frac{n}{2}}$$

in which h , ν , A , and E_g represents Planck constant, light frequency, proportionality constant and band gap, respectively, while n depends on the nature of transition in a semiconductor. Values of 1, 3, 4 and 6 for n correspond to allowed direct, forbidden direct, allowed indirect, and forbidden indirect transitions, respectively. The values of E_g were determined from the plot of $(F(R)hv)^{2/n}$ against $h\nu$ and corresponded to the intercept of the extrapolated linear portion of the plot near the band edge with the $h\nu$ axis. Pt/TiO₂ samples were treated as the semiconductors with allowed indirect transition. The values of E_g were thus determined from the plot of $(F(R)hv)^{1/2}$ against $h\nu$. The figure of Fig. S2b has been corrected.

Comment 6. *“The time-resolved fluorescence decay spectra of various Pt/TiO₂ should be provided.”*

Reply: The photoluminescence (PL) spectra of the Pt/TiO₂ catalysts at an excitation wavelength of 365 nm have been added as Figure S8 in the revised manuscript (Figure R4). The emission peak around 400 nm arises due to self-trapped excitons (STE) localized on TiO₆ octahedra⁴ or to an oxygen vacancy with two trapped electrons⁵. The STE can be created by direct recombination of a trapped electron on the lattice site with a hole or the by indirect recombination assisted by an oxygen vacancy. The emission signals around 468 nm and 552 nm can be assigned to the charge-transfer from Ti³⁺ to oxygen anion in a TiO₆ octahedron associated with oxygen vacancies⁶⁻⁸. The emission around 400 nm may originate from oxygen vacancies that form shallow traps near the surface of the nanocrystals, while the emission at 552 nm is associated with oxygen vacancies that form deep traps. It is seen that there exists a direct correlation between the intensity ratio of the emission peaks from shallow and deep traps and the photocatalytic activity of the samples. The emission intensity at 400 nm is found to be the maximum for the sample 1% Pt/TiO₂(A) which exhibited the highest photocatalytic activity. From our analysis, it can be inferred that the shallow trap states would have helped in the effective separation of the charge carriers to facilitate efficient

photocatalytic activity. Deep trap centers, on the other hand, favor recombination of charge carriers and hinder the photocatalytic activity. The emission intensity of 1% Pt/TiO₂(R) is found to be the maximum at 552 nm while minimum at 416 nm, which exhibited the lowest photocatalytic activity.

Fig R4. PL spectra of various Pt/TiO₂ catalysts.

Comment 9. “The quality of the manuscript should be carefully improved. For example, the figure is not consistent with the text (please ref to, line 122 (where Figure 1A should be changed to Figure 2A), line 126 and line 134.), and the reference format is also incorrect; The format for “mL” should be consistent, please refer to line 103 and 151.”

Reply: Thanks for the advice and we have checked the revised manuscript times to correct the mistakes.

Referee #2

Dear authors,

this is an interesting manuscript that reports a photocatalytic conversion of Acetonitrile to Succinonitrile over titania-supported Pt catalysts. The work was carefully conducted and the manuscript is well written. However, it seems that this manuscript is not appealing enough to the broad readership of Nature Communications because the

Succinonitrile market is small. It is not acceptable to claim that "the annual demand of succinonitrile is growing rapidly" based on a resource that was published 15 years back (ref.3, Applied Microbiology and Biotechnology 2007, 75 (4), 751-762.). This reviewer recommends the authors to redirect their work to some of the chemical engineering journals.

Reply: In this paper, we have demonstrated that the photocatalytic method can provide environmentally friendly, simple and atom economical synthesis of industrial important value-added chemical based on abundant renewable energy. It has also demonstrated that the $\cdot\text{OH}$ radical initiated coupling of small molecule can highly selectively produce terminal bifunctional molecules, which is scientific important. Additionally, DAB, the major downstream product of succinonitrile is a very important chemical with a market valued 124 million dollars per year. The annual growth rate is about 4%. We added one more reference in the introduction to claim the importance of this work (<https://www.marketresearch.com/QYResearch-Group-v3531/Global-Diaminobutane-Research-12540796/>)

Referee #3 The reviewer 3 thought our system is remarkably efficient, and recommend publication after consideration of some points.

Comment 1. *"In figure 2D, I think the reaction is limited by photons, not the amount of catalyst. Can some estimate be made of the quantum efficiency of the reaction. i.e., moles of product/mole of photons?"*

Reply: AQY calculation methods.

In generally, producing one $\cdot\text{CH}_2\text{CN}$ radical from CH_3CN needs one hole with one electron consumed at the same time (e.g., one SN molecule needs two electrons). Then the AQY could be calculated according to the following Equation:

$$\begin{aligned}\text{AQY}(\%) &= \frac{\text{number of reacted electrons}}{\text{number of incident photons}} \times 100\% \\ &= \frac{\text{number of evolved SN molecules} \times 2}{\text{number of incident photons}} \times 100\%\end{aligned}$$

The total light intensity incident (305 mW/cm^2) at the sample position is measured

by using CEL-NP2000-2A flux meter. Assuming uniform intensity distribution of the lamp, a correction for the difference in the area of the sensor of lux meter and the reactor surface area is evaluated.

AQY calculation

Number of incident photons:

$$N = \frac{I \times A \times \lambda \times t}{h \times c} = \frac{3.05 \times 10^3 \times 1.77 \times 10^{-4} \times 365 \times 10^{-9} \times 3600}{6.626 \times 10^{-34} \times 3 \times 10^8} = 3.57 \times 10^{21}$$

I: optical power density ($\text{W} \cdot \text{m}^{-2}$)

A: incident illumination area (m^2)

λ : wavelength of incident light (m)

t: time (s)

H: Planck constant (6.626×10^{-34} J·s)

C: the speed of light (3×10^8 m·s⁻¹)

$$\begin{aligned} \text{AQY}(\%) &= \frac{\text{number of evolved SN molecules} \times 2}{N} \times 100\% \\ &= \frac{N_A \times n \times 2}{N} \times 100\% = \frac{6.02 \times 10^{23} \times 1.31 \times 10^{-4} \times 2}{3.57 \times 10^{21}} \times 100\% = 4.4\% \end{aligned}$$

N_A : Avogadro's constant (6.02×10^{23} mol⁻¹)

n: moles of SN per hour

The AQY data has been updated in the table S1 and S2 of the revised manuscript. The experimental procedure was added into the supporting information.

We agreed with the reviewer that the flux of photon influenced the reaction rate. When lower photon flux was used, a dramatic decrease was observed on the coupling possibility.

Table R2. Evaluation of the radiation intensities on the catalytic performances

Entry	Radiation intensities	SN Formation rate (mmol g _{cat} ⁻¹ h ⁻¹)
1	1 w	1.22
2	2 w	1.81
3	5 w	3.19
4	7 w	4.76

Comment 2. “p.6, top, succinonitrile is produced at a rate of 6.55 mmol/g/h, whereas H₂ is produced at a rate of 9.76 mmol/g/h. These rates should be equal. Why/how is the excess of hydrogen produced?”

Reply: First of all, CO₂ can be dissolved in the solution (the saturation concentration in acetonitrile is 6 times higher than pure water). Therefore, the GC estimation of CO₂ may be lower than the generation. This is the major reason for the difference between the H₂ production and SN production rates. Secondly, there are trace amount of colored organic impurities formed due to the over oxidation, which may also have minor influence on the electron balance. The error of GC quantification also contributed to the unbalanced results. The phenomenon is common in photocatalytic reactions generating CO₂. For example, Wang et al. performed the photocatalytic reforming of biomass to produce hydrogen. The H₂/CO₂ ratio was determined at 9/1 by GC. (J. Am. Chem. Soc. 2021, 143, 6533–6541).

Even though all the extra H₂ is related with the dissolved CO₂, the selectivity of SN based on carbon balance was still quite high. For example, for the case mentioned by the reviewer, the maximum CO₂ production rate would be 1.98 mmol/g/h based on the hydrogen formation rate. In this situation, the SN selectivity would only drop from the 97.5 to 92.0%.

Comment 3. “the mechanistic studies of this heterogeneous system are suggestive, but not proof, of that mechanism involves CH₂CN radicals. The observation of spin-trapped radicals is perhaps the best indication of CH₂CN radicals. However, the isotope effect studies are not what I expected. Is there any indication of why the isotope effects (kH/kD) are not multiplicative? (1.6 x 4.3 = 6.9, not 13.1).”

Reply: We did the kinetic isotope effect experiment three times and found the reaction rate using D₂O and CD₃CN should was 0.85±0.05 mmol/(gcat*h). Therefore, the kH/kD value equals to 6.8~7.7.

Comment 4. “in the SI, p.5, how was the number of catalyst active centers determined in Eq 2?”

Reply: The activity reported in the manuscript is the Pt normalized activity, which means all the Pt loaded over the support was calculated. The loading of Pt was determined by the ICP-OES method.

Comment 5. “in the SI, p.16, Table S4. Why does the rate fall off with reaction time? This is seen for SN, H₂ CO₂ and AM. It would be better to give amounts (in mmol) of products, not just rates which are presumably derived from the observed amounts.”

Reply: The reasons of the deactivation of the photocatalyst are complicated and we have investigated the problem but can only provide some possibilities. First of all, we suspected the accumulation of the products and byproducts may cause the deactivation. However, after adding SN (10 mmol/L), H₂ (0.04 mmol) and AM (10 mmol/L) into the system at the beginning of the reaction, no obvious deactivation was observed (see Table R3). Then we discovered that loading of Pt dropped from 0.91 to 0.75% after reaction, which might be a possible reason for the deactivation. After the detailed analysis of the reaction medium after reaction, we found there was trace amount of cyanide generated (Table R4) with time, which may lead to the leaching of the noble metal center. They may be formed via the following route.

We are now working on how to prevent the generation of the cyanides and improve the stability of the catalysts. Meanwhile, there might be other potential reasons for the loss of activity which is currently beyond our understandings.

The Table S4 have been updated according to the reviewer’s suggestion.

Table R3. SN formation rate of different added substance

Entry	Added substance	SN Formation rate (mmol g _{cat} ⁻¹ h ⁻¹)
1	SN (10 mmol/L)	6.26
2	H ₂ (0.04 mmol)	6.78

3	AM (10 mmol/L)	6.19
---	----------------	------

Table R4. Change of CN⁻ concentration with reaction time

Entry	Reaction time (h)	Concentration of CN ⁻ (ug/mL)
1	0	0
2	2	57.31
3	4	66.31
4	6	76.16
5	8	83.03
6	10	91.88
7	12	95.17

Fig R5. UV-vis spectra of CN⁻ concentration in different reaction time.

Comment 6. Other corrections

p.1, line 23 should be “Under optimized conditions”

p.2, line 1 should be “under mild conditions”

p.2, line 2 from bottom should be “in this atom economic pathway, the highly toxic hydrogen cyanide is used”

p.3, line 50 should be “assisted with an organometallic complex”

p.3, line 54 should be “of the sp³ bond”

p.3, line 55 should be “large-scale production.”

p.3, line 61 should be “under ambient conditions.”

p.4, line 81 should be “The resulting catalysts”

p.4, line 86 should be “numbers of each of the catalysts”

p.4, line 5, Fig. 1 should be “The particle size distribution”

p.5, line 104 should be “reaction medium and reactant.”

p.6, line 130 should be “CH₃CN was transformed”

p.7, line 143 should be “have changed with a reduced amount of catalyst.”

p.8, line 159 should be “When AgNO₃ was added”

p.8, line 161 should be “significant amounts of”

*p.8, line 162 should be “with rates of 15.56 and 9.75 mmol/(g*h), respectively.”*

p.8, line 172 should be “Kinetic isotope effect experiments suggested”

p.8, line 174 should be “The hydrogen atom in the H₂ originated from” also, I would say that water is acting as a promoter, not a co-catalyst.

p.8, line 178 should be “assisted by radical trap reactions were performed”

p.10, line 212 should be “evolution, exhibiting high side reaction selectivity.” p.10, line 213 should be “to achieve a high yield”

p.10, line 219 should be “from the catalyst surface.”

p.S3, 2.2, give the amounts of TiO₂ powder and metal precursor.

p.S4, line 61, should be “Kapton tape”

p.S11, Fig S6 should be “Product analysis”

p.S12, Fig S7, the scale for the NMR spectra are too small to read.

p.S17, Table S5, footnote should be “those shown in Table S2”

Reply: We appreciated for the valuable suggestions for us to improve the quality and readership of this paper. Corrections have been made in the revised manuscript. We have also checked the revised paper times before submission.

Reference

- (1) Shi, J. W.; Ye, J. H.; Zhou, Z. H. et al. Hydrothermal Synthesis of Na_{0.5}La_{0.5}TiO₃-LaCrO₃ Solid-Solution Single-Crystal Nanocubes for Visible-Light-Driven Photocatalytic H₂ Evolution. *Chem. Eur. J.* **2011**, 17, 7858-7867.
- (2) Anpo, M.; Tomonari, M.; Fox, M. A., In situ photoluminescence of TiO₂ as a probe of photocatalytic reactions. *J. Phys. Chem.* **1989**, 93, 7300-7302.
- (3) Carey, J. J.; McKenna, K. P., Screening doping strategies to mitigate electron trapping at anatase TiO₂ surfaces. *J. Phys. Chem. C.* **2019**, 123, 22358-22367.
- (4) S. Zhou, A.K. Ray, Kinetic studies for photocatalytic degradation of Eosin B on a thin film of titanium dioxide, *Ind. Eng. Chem. Res.* **2003**, 42, 6020-6033.
- (5) T. Sekiya, K. Ichimura, M. Igarashi, S. Kurita, Absorption spectra of anatase TiO₂ single crystals heat-treated under oxygen atmosphere, *J. Phys. Chem. Solid.* 61 (2000) 1237–1242
- (6) H. Zhang, M. Zhou, Q. Fu, B. Lei, W. Lin, H. Guo, M. Wu, Y. Lei, Observation of defect state in highly ordered titanium dioxide nanotube arrays, *Nanotechnology* 25 (2014) 275603
- (7) L. V Saraf, S.I. Patil, S.B. Ogale, S.R. Sainkar, S.T. Kshirsager, Synthesis of nanophase TiO₂ by ion beam sputtering and cold condensation technique, *Int. J. Mod. Phys. B* 12 (1998) 2635–2647
- (8) Y. Lei, L.D. Zhang, G.W. Meng, G.H. Li, X.Y. Zhang, C.H. Liang, W. Chen, S. X. Wang, Preparation and photoluminescence of highly ordered TiO₂ nanowire arrays, *Appl. Phys. Lett.* 78 (2001) 1125–1127

REVIEWERS' COMMENTS

Reviewer #1 (Remarks to the Author):

In the revised manuscript, most of the reviewer's concerns have been well-addressed. The results in table R1 could be helpful for the readers to understand the reaction-scope as well as the mechanism insights, thus Table R1 is recommended to be included in SI. Although the scale-up experiment is not very satisfied, the potential has been initially proved, which could be further optimized from chemical engineering view-points. In summary, the revised manuscript has been obviously improved, thus, publication of this version on Nature Communications could be now recommended.

Reviewer #3 (Remarks to the Author):

2022Nat-Yao

The authors have adequately responded to the issues that I raised. In particular, the isotope effects in Figure 3 are now multiplicative. Also, in addressing the comments about commercialability, the authors examined a similar reaction with methanol. Ethylene glycol is produced. This product is of industrial importance, so it would be interesting to see a future investigation of this reaction.